# The Emerging, Multifaceted Role of WTAP in Cancer and Cancer Therapeutics

**DOI:** 10.3390/cancers15113053

**Published:** 2023-06-04

**Authors:** Guomin Ju, Jiangchu Lei, Shuqi Cai, Siyuan Liu, Xinjia Yin, Chuanhui Peng

**Affiliations:** 1Division of Hepatobiliary and Pancreatic Surgery, Department of Surgery, The First Affiliated Hospital, Zhejiang University School of Medicine, Hangzhou 310003, China; ju_guomin@zju.edu.cn (G.J.); joshua_rae0802@zju.edu.cn (J.L.); caisq@zju.edu.cn (S.C.); 3190102278@zju.edu.cn (S.L.); 3190102269@zju.edu.cn (X.Y.); 2NHC Key Laboratory of Combined Multi-Organ Transplantation, Hangzhou 310003, China; 3Key Laboratory of the Diagnosis and Treatment of Organ Transplantation, Research Unit of Collaborative Diagnosis and Treatment for Hepatobiliary and Pancreatic Cancer, Chinese Academy of Medical Sciences (2019RU019), Hangzhou 310003, China; 4Key Laboratory of Organ Transplantation, Research Center for Diagnosis and Treatment of Hepatobiliary Diseases, Hangzhou 310003, China

**Keywords:** cancer, WTAP, m^6^A, methylation

## Abstract

**Simple Summary:**

N6-methyladenosine (m^6^A) modification is among the most common and conservative RNA modifications in eukaryotes. The deposition of m^6^A on RNA is carried out by methyltransferases. As a regulatory subunit of methyltransferase, WT1-associated protein (WTAP) has gradually received attention in recent years. WTAP has been found to be expressed abnormally in a large number of cancers and affects cancer progression and prognosis. In this review, we propose a new perspective on the impact of WTAP on the occurrence and development of cancer, summarize the functional classification of WTAP in cancer, and envision potential therapeutic prospects.

**Abstract:**

Cancer is a grave and persistent illness, with the rates of both its occurrence and death toll increasing at an alarming pace. N6-methyladenosine (m^6^A), the most prevalent mRNA modification in eukaryotic organisms, is catalyzed by methyltransferases and has a significant impact on various aspects of cancer progression. WT1-associated protein (WTAP) is a crucial component of the m^6^A methyltransferase complex, catalyzing m^6^A methylation on RNA. It has been demonstrated to participate in numerous cellular pathophysiological processes, including X chromosome inactivation, cell proliferation, cell cycle regulation, and alternative splicing. A better understanding of the role of WTAP in cancer may render it a reliable factor for early diagnosis and prognosis, as well as a key therapeutic target for cancer treatment. It has been found that WTAP is closely related to tumor cell cycle regulation, metabolic regulation, autophagy, tumor immunity, ferroptosis, epithelial mesenchymal transformation (EMT), and drug resistance. In this review, we will focus on the latest advances in the biological functions of WTAP in cancer, and explore the prospects of its application in clinical diagnosis and therapy.

## 1. Introduction

Cancer, a worldwide public health problem, is among the main causes of death in every country in the 21st century. An analysis of global cancer data from 1990 to 2019 showed that 23 million people had cancer in 2019, more than twice the number in 1990 [1]. Over the years, it has been shown that multiple mechanisms could induce the occurrence of cancer, including chromosome translocation, and gene deletion, amplification, and mutation [2,3]. Nevertheless, there is increasing evidence that epigenetic transcription also plays an important role in the occurrence and development of cancer [4,5,6]. Epitranscriptomics involves a wide range of aspects, including chromatin remodeling, DNA methylation, histone modification, cancer immunity, and non-coding RNA regulation, among which RNA methylation is crucial [7].

Scientists have successively discovered various RNA methylation modifications, including N6-methyladenosine (m^1^A) [8], 5-methylcytosine (m^5^C) [9], and N6-methyladenosine (m^6^A) [10]. M^6^A modification is considered to be the most common internal modification in eukaryotic messenger RNA (mRNA) [10]. Since it was first identified in 1974, m^6^A modification has been proven to exist in the mRNAs of over 7000 genes and in over 300 non-coding RNAs [11]. On average, there are 1–2 m^6^A residues per 1000 nucleotides, with these residues being enriched in the 3’UTR and located within the consensus sequence RRACH (R = G or A, H = A, C, or U) [12,13,14]. M^6^A modification could regulate the function and expression of RNA by regulating various stages of the RNA cycle, such as RNA processing, transportation, localization, or translation [15]. Similar to DNA methylation, m^6^A modification in mammalian cells is reversible. Three classes of enzymes control the modification of m^6^A: writers, readers, and erasers [16]. The writers include various m^6^A methyltransferase proteins (METTL3, METTL14, WTAP, KIAA1429, VIRMA, RBM15, RBM15B, METTL16), which together form the mRNA methyltransferase complex and are responsible for adding a methyl group to specific adenines [17]. The readers recognize modified adenine and perform specific functions, while the erasers perform demethylation. The m^6^A recognition proteins (YTHDF1, YTHDF2, YTHDF3, YTHDC1, YTHDC2, elF3, HNRNPA2/B1, HNRNPC/G, IGFBPs) and the demethylase proteins (FTO, ALKBH5) are the other two types of enzymes responsible for the functional operation or remodeling of this epigenetic modification (Figure 1) [17,18].

WTAP is the regulatory subunit of methyltransferase. In the absence of WTAP, the methyltransferase’s RNA binding capacity is markedly decreased [19]. It has been proven that the removal of WTAP gene leads to embryonic lethality [20]. Moreover, an increasing amount of evidence suggests that WTAP contributes to the aggressive characteristics of numerous cancers. In this article, we will focus on the latest research on the biological function and regulatory mechanism of WTAP in cancer, and explore the prospects and possible future research directions of WTAP in clinical applications.

## 2. M^6^A Methyltransferase

M^6^A methyltransferase catalyzes the modification of m^6^A methylation of mRNA and consists of at least eight components, including METTL3, METTL14, WTAP, zinc finger CCCH-type containing 13 (ZC3H13), METTL16, vir-like m^6^A methyltransferase-associated (VIRMA), RNA-binding motif protein 15 (RBM15), and RNA-binding motif protein 15B (RBM15B). METTL3, the first methyltransferase to be identified, holds a position of utmost significance as a constituent member of the methyltransferase complex, as it displays catalytic activity as an N6-methyltransferase [21]. METTL14 forms heterodimers with METTL3 in a 1:1 ratio, playing a crucial role in substrate recognition [22]. Despite it lacking catalytic activity, WTAP ensures the METTL3-METTL14 heterodimer is localized to nuclear speckles, thereby facilitating its catalytic activity [22]. Meanwhile, ZC3H13 enhances m^6^A modification by anchoring WTAP in the nucleus [23]. METTL16 could mediate m^6^A methylation of precursor mRNA (pre mRNA), non-coding RNA, and U6 snRNA, while VIRMA recruits METTL3 and METTL14 to regulate regioselective methylation, thereby inducing mRNA splicing and RNA processing [24,25]. Furthermore, RBM15 combines with RBM15B to bind the MTC and recruit it to specific sites in the transcript [26]. Moreover, there exist additional m^6^A methyltransferases that have been identified, including METTL4, METTL5, and zinc finger CCHC domain-containing protein 4 (ZCCHC4) [27,28,29,30].

## 3. Function and Role of WTAP

WTAP, as the partner of Wilms’ tumor 1 (WT1), was first isolated and identified through a yeast two hybrid system in 2000 [31]. Dysregulation of WTAP was associated with certain substantive cellular processes, such as X-chromosome inactivation [32], cell proliferation [19], cell cycle regulation [20], and selective splicing [33]. In recent years, WTAP has received increasing attention, and a large number of reports showed that WTAP played an important role in various types of cancer, such as liver cancer, esophageal cancer, breast cancer, bladder cancer, lung cancer, and lymphoma [34]. Recent studies have shown that abnormal expression of WTAP is closely related to various pathophysiological events in cancer, including the cell cycle, metabolic vulnerabilities, autophagy, immune response, ferroptosis, epithelial mesenchymal transition (EMT), and drug resistance (Figure 2) (Table 1).

### 3.1. WTAP and the Cell Cycle

A disordered cell cycle is the basis for uncontrolled tumor cell proliferation characterized by malignant phenotypes [62]. Each pathway that limits the normal cell proliferation response is interfered with in most cancers. Bioinformatics analysis found that WTAP was associated with the cell cycle of hepatocellular carcinoma [36]. Specifically, WTAP knockdown induced cell cycle arrest in the G2/M phase via upregulating the ETS1-P21/P27 axis [37]. It was reported that in nasopharyngeal carcinoma (NPC), decreased expression of WTAP induced G1 phase cell cycle arrest and stimulated apoptosis [35]. In acute myeloid leukemia (AML), downregulation of WTAP induced G1/S phase stagnancy, resulting in decreased proliferation of tumor cells, and made cells more prone to apoptosis [41]. Interestingly, in vitro experiments also found that WTAP knockdown cells were more fragile than control cells after simultaneous treatment with daunorubicin, indicating that WTAP knockdown caused a decrease in drug resistance [41]. Unlike cancer cells, in vascular smooth muscle cells (SMCs), restrained WTAP increased SMC proliferation via promoting DNA synthesis and G1/S phase transition, while reducing cell apoptosis [63].

The progression through the cell cycle is mainly regulated by cyclins and the cyclin dependent kinase (CDK) family of serine/threonine kinases [64]. On one hand, quite a few studies have found that WTAP had a positive effect on the cell cycle progression of normal cells through the mediation of cyclins and the CDK family. In umbilical vein endothelial cells, WTAP knockdown prevented G2/M phase transition via reducing the stability of cyclin A2 mRNA [20]. In addition, cyclins B1, B2, and CDC20 also decreased to a certain extent [20]. Previous studies have shown that WTAP promoted G2/M transformation of keratinocytes, which may be related to cyclin A2 and CDK2, but the specific mechanism needed further experiments [65]. Furthermore, WTAP actively regulated the process of adipogenesis by controlling the cell cycle transition of mitotic clonal expansion (MCE), which is also related to cyclin A2 [66]. On the other hand, in cancer cells, WTAP played a role in enhancing CDK2 protein expression by binding to the 3’-UTR of the CDK2 transcript to stable the CDK2 transcript [38]. In renal cancer, downregulation of WTAP not only caused downregulation of CDK2, but also downregulation of other G1/S transition regulators, including CDK4, CDK6, and CCND1, leading to severe G1/S arrest [40]. In endometrial cancer, WTAP deficiency led to increased expression of BCL2-associated X (BAX) and cleave-PARP, decreased expression of Myeloid Leukemin-1 (Mcl-1), and induction of G2/M phase arrest of the cell cycle [39].

### 3.2. WTAP and Metabolic Vulnerabilities

The rapid proliferation of cancer cells requires a large number of cellular catabolism and anabolism to meet the structural and energy demands [67]. Changes in major metabolic pathways [68], particularly those involving glucose and lipids [69], are among the most significant metabolic characteristics spanning different cancer types [70]. In addition, it is now clear that different carcinogenic drivers induce different metabolic phenotypic alterations in cancer cells.

In contrast to normal differentiated cells that primarily rely on mitochondrial oxidative phosphorylation, most cancer cells rely on aerobic glycolysis to generate the energy essential for cellular processes (a phenomenon termed ‘the Warburg effect’) [71]. Hexokinase 2 (HK2) plays a major role in intracellular glucose utilization. Induction of HK2 in most tumor cells contributed to their metabolic predisposition to aerobic glycolysis. Additionally, its genetic knockdown inhibited malignant growth in mouse models [72]. There was convincing evidence that WTAP could widely affect the Warburg effect by targeting HK2 in various tumor cells. In diffuse large B cell lymphoma (DLBCL), WTAP increases the expression of its key target gene HK2 by elevating the level of HK2 m^6^A to promote DLBCL [43]. WTAP, an oncogene, is overexpressed in gastric cancer cells [44]. In vitro experiments have shown that WTAP promoted tumor cell proliferation and glycolysis (including glucose uptake, lactic acid production, and extracellular acidification rate). Mechanistically, WTAP stabilized HK2 mRNA, and enhanced glucose uptake of gastric cancer cells [44]. In ovarian cancer, WTAP interacted with DGCR8, a key chip protein, to affect the expression of microRNA-200 (miR-200) by regulating m^6^A. Then, miR-200 further regulated HK2 positively, and significantly affected the Warburg effect [45]. In colon adenocarcinoma (COAD), Zhang et al. found that knocking down WTAP can also inhibit glucose consumption and lactate production, thereby inhibiting tumor progression [42]. Further analysis has shown that WTAP mediated the stability of Forkhead Box P3 (FOXP3) in an m^6^A-dependent manner to promote SMARCE1 transcriptional activation and ultimately enhance glycolysis in colon adenocarcinoma [42]. In addition, WTAP was regulated by Caprin-1 in esophageal cancer [46] and extracellular regulated protein kinase 1 (ERK1) and extracellular regulated protein kinase 2 (ERK2) in breast cancer [47] to promote the glycolytic activity of tumor cells. On the other hand, WTAP modified and enhanced the expression of insulin-secretion-related genes and specific transcription factors in pancreatic islets beta cells by increasing the level of METTL3 protein [73]. Therefore, WTAP could become a promising therapeutic target for certain tumors through the glycolysis pathway.

Lipid metabolism is an important cellular process that converts nutrients into metabolic intermediates for membrane biosynthesis, energy storage, and signal molecule production. Changes in lipid metabolism are particularly prominent in the metabolic changes of cancer. Additionally, enhancing lipid uptake or synthesis contributes to tumor formation and rapid growth of cancer cells [74]. Several studies have shown that lipogenesis was crucial for tumor growth [75,76,77]. WTAP, METTL3, and METTL14 formed a complex that actively controlled lipogenesis by promoting cell cycle transitions in mitotic clonal amplification (MCE) during lipogenesis [66]. In vitro experiments have found that hepatic deletion of WTAP inhibits the upregulation of cyclin A2 during MCE, leading to cell cycle arrest and impaired adipogenesis [66]. Recently, Li et al. revealed that hepatic deletion of WTAP could lead to non-alcoholic steatohepatitis (NASH), which is one of the most common potential risk factors for hepatocelluar carcinoma (HCC) [78,79]. Mechanistically, liver conditional knockout of WTAP enhanced lipolysis in adipose tissue by inducing the expression and secretion of IGFBP1, and increased the expression of CD36 and CCL2, thereby enhancing liver free fatty acids (FFA) uptake and inflammation [78]. In short, these findings provide new ideas for the study of adipogenesis in tumors.

### 3.3. WTAP and Autophagy

Autophagy is a highly conserved process of cellular decomposition and metabolism that can be divided into three types: macro-autophagy, micro-autophagy, and chaperone-mediated autophagy (CMA) [80]. Autophagy plays a dual role in the development of cancer, but there is sufficient evidence that autophagy, as a tumor suppressor, inhibits the occurrence and development of liver tumors. In vitro experiments have found that downregulation of WTAP promoted the conversion of LC3-I to LC3-II (indicating autophagy formation), and inhibited the proliferation and metastasis of liver cancer cells by promoting autophagy [48]. Further analysis showed that WTAP targeted liver kinase B1 (LKB1), which in turn mediated the phosphorylation of AMP-activated protein kinase (AMPK) through m^6^A, thereby promoting the progression of hepatocellular carcinoma [48]. Overexpression of WTAP could protect cells from autophagic death. Autophagy in liver cancer cells can reduce inflammation and cell damage through its organelle quality and protein control function, and then prevent the initiation and development of tumors [81]. In short, inducing autophagy in liver tumors by targeting WTAP may be an effective method for developing new promising therapies.

### 3.4. WTAP and Immune Response

Tumor progression is closely related to immune infiltration in the tumor microenvironment. The immune system is regulated by the interaction between cells mediated via cytokines, and shows a dynamic equilibrium between tumor infiltration and peripheral cisterns [82,83,84]. Wang et al. discovered that overexpression of WTAP in Treg cells positively promoted Treg cell differentiation and enhanced the inhibition of immature T cells mediated by Treg cells [85]. Further mechanism studies indicated that FOXO1 was the downstream target of WTAP, and WTAP upregulated FOXO1 protein levels via the enhancing m^6^A modification of FOXO1 mRNA [85]. In addition, mice with WTAP deficiency of CD4^+^T cells exhibited a phenotype of impaired thymocyte growth and peripheral T cell reduction [86]. Further analysis showed that T cell receptor (TCR)-induced T cell expansion was negatively affected by WTAP deficiency. Interestingly, WTAP deficiency in combination with TCR stimulation strongly induced apoptosis, but in the absence of TCR stimulation, depletion of WTAP did not alter the activity of initial CD4^+^T cells [86]. Therefore, it was found that WTAP has a very close relationship with the immune system. However, there is currently a significant lack of research on the impact of WTAP on tumor immunity. In cancer cells, bioinformatic analysis demonstrated that overexpression of WTAP was significantly positively correlated with CD4^+^T cells, CD8^+^T cells, B cells, cancer associated fibroblasts, bone marrow dendritic cells, neutrophils, and macrophages, while it was significantly negatively correlated with Treg cells [36,49]. Based on the previous studies mentioned above, WTAP may play a potential role in tumor immunity, which requires further research.

### 3.5. WTAP and Ferroptosis

Promoting cancer cell death is an effective cancer treatment method. The common types of cell death currently include necroptosis, pyroptosis, ferroptosis, parthanatos, immunogenic cell death (ICD), lysosome-dependent cell death (LCD), necrotic cell death (NCD), and autophagy-dependent cell death [87]. However, current research on WTAP and tumor cell death is limited to ferroptosis [50,51]. Ferroptosis is an iron-dependent programmed cell death characterized by a significant accumulation of reactive oxygen species (ROS) and mortal lipids that is different from autophagy, necrosis, and apoptosis [88,89,90]. Targeting ferroptosis will provide a new idea for treating cancers that are difficult to deal with using traditional therapies. On one hand, ferroptosis contributed to cancer development by regulating some tumor suppressors. On the other hand, ferroptosis weakened some types of cancer cells and these may be developed as a potential therapeutic target [91]. In hepatocellular carcinoma, WTAP knockdown reduced the m^6^A level of circCMTM3, inhibited the expression of circCMTM3, and increased ferroptosis-related biomarkers (MAD, iron, and Fe^2+^), suggesting that ferroptosis was induced [50]. Specifically, circCMTM3 regulated by WTAP bound with IGF2BP1 to regulate the expression of PARK7, which in turn affected ferroptosis in HCC [50]. WTAP has been found to promote the proliferation of bladder cancer cells and inhibit the ferroptosis induced by erastin [51]. Mechanism studies have shown that WTAP enhances the modification of NRF2 mRNA 3’-UTR and increased the expression of NRF2 through YTHDF1, thereby inhibiting ferroptosis. The m^6^A binding motif is AACCA [51]. These findings provide insights on how to target WTAP in combination with ferroptosis for cancer therapy. Certainly, other cell death modes are also worth exploring.

### 3.6. WTAP and EMT

EMT, the process by which epithelial cells acquire mesenchymal characteristics, endows cells with the ability to migrate and invade. EMT is important for various tumor biological functions, including tumor initiation, tumor cell migration, malignant progression, metastasis, blood infiltration, and resistance to treatment [92,93,94]. It was reported that knockdown of WTAP in liver cancer cells significantly inhibited the proliferation, migration, and invasion of tumor cells [53]. Mechanism analysis showed that miR-139-5p-mediated WTAP regulated HCC progression by controlling EMT. Specifically, the study has discovered that knockdown of WTAP decreased several mesenchymal markers including Slug, Snail, and N-cadherin, and increased E-cadherin expression [53]. The imbalance of EMT modulators such as E-cadherin, N-cadherin, Slug, Snail, and vimentin is closely related to tumor invasion and adverse clinical outcomes [95]. Similar results have also been reported by Yu et al. [54]. Dysregulation of WTAP in ovarian cancer reduced the expression of vimentin and increased the expression of E-cadherin, thereby inhibiting migration [54]. In colon cancer, WTAP upregulated the expression of Snai1 by increasing m^6^A levels, thereby promoting lung metastasis [52]. Mechanistically, TTC22 was upstream of WTAP and increased the level of WTAP protein by promoting the binding of 60S ribosomal protein L4 (RPL4) to WTAP mRNA [52]. Epidermal growth factor (EGFR) played key roles in the proliferation and migration of cancer cells [96]. However, when EGFR signal transduction was altered, it became the primary coordinator of epithelial transformation. Jin et al. found that overexpression of WTAP enhanced EGFR phosphorylation without affecting total EGFR to promote the migration and invasion of glioblastoma cells [55].

### 3.7. WTAP and Drug Resistance

Chemotherapy is one of the most effective measures for treating cancer, and together with surgery and radiotherapy, these are known as the three major treatment measures for cancer. However, the effectiveness of anti-tumor chemotherapy drugs is often limited by tumor resistance. Cisplatin is a common anticancer drug. The most acceptable mechanism is to inhibit cell division and interact with purine bases on DNA to cause DNA damage, leading to cell apoptosis [97]. WTAP depletion has been shown to promote cisplatin resistance in endometrial carcinoma cells by inducing cell cycle arrest [39]. The Wnt/β-catenin pathway is a downstream target of WTAP [39]. Similarly, WTAP knockdown has been proven to inhibit cisplatin resistance in natural killer/T cell lymphoma (NKTCL) cells [59]. In bladder cancer, WTAP can also modulate cisplatin sensitivity by enhancing the expression of TNF Alpha Induced Protein 3 (TNFAIP3) [61]. In esophageal cancer, WTAP in hypoxic environments is upregulated by lncRNA EMS and promotes cisplatin resistance in cancer cells [60]. Etoposide is often used in combination with cisplatin, and the therapeutic mechanism is also related to the cell cycle [98]. In vitro experiments have shown that knockdown of WTAP reduces etoposide resistance in diffuse large B cell lymphoma cells [57]. Anthracyclic drugs (such as doxorubicin and daunorubicin) function by promoting the formation of reactive oxygen species (ROS) and inhibiting DNA and RNA synthesis. In breast cancer, the WTAP-induced DLGAP1-AS1 axis inhibits the sensitivity of tumor cells to doxorubicin [58]. Moreover, the effect of WTAP on daunorubicin resistance is achieved by influencing tumor cell proliferation, and regulating the G1/S phase transition and differentiation [41]. In addition, WTAP has also become a regulator of resistance to pyrimidine antimetabolic chemotherapy drugs. WTAP can activate the Fak-Src-GRB2-Erk1/2 and Fak-PI3K-AKT signaling pathways to restrain the chemosensitivity of pancreatic cancer cells to gemcitabine, and Fak inhibitors have been shown to be able to rescue high WTAP-mediated chemo-tolerance to gemcitabine in pancreatic cancer [56]. To sum up, targeting WTAP and its upstream and downstream targets may be a feasible strategy for suppressing drug resistance and cancer treatment.

## 4. Potential Clinical Applications of WTAP

Based on the multiple functions of WTAP, targeting WTAP may become a new perspective for cancer therapy. In most cancers studied, WTAP is almost an oncogene, closely related to the occurrence and progression of tumors. Specifically, WTAP was upregulated in various types of cancer, such as liver cancer, esophageal cancer, AML, and osteosarcoma [37,41,49,99]. Moreover, the upregulation of WTAP promoted tumor proliferation and metastasis, usually indicating poor prognosis, and was an independent prognostic risk factor for cancer. In addition, Wang et al. found that WTAP was more highly expressed in poorly differentiated colorectal cancer (CRC) tissue, suggesting that WTAP may be a malignant feature that is positively correlated with the degree of malignancy of the tumor [100]. Therefore, targeted WTAP could identify abnormalities and serve as a diagnostic or prognostic marker for different forms of cancers. Given the carcinogenic effect of and the involvement of various pathophysiological events, WTAP may be a promising target for cancer treatment. However, no inhibitor of WTAP has been found. Due to the crucial function of WTAP in cancer progression, it is urgent to design and develop effective WTAP inhibitors. Drug resistance has always been an obstacle during clinical chemotherapy, and WTAP plays a prominent role in regulating chemical resistance. The reduced WTAP expression heightened the chemosensitivity of different cancers to diverse medications, comprising cisplatin, gemcitabine, and anthracyclines [39,41,56]. The combination of WTAP inhibitors and chemotherapy has great therapeutic potential and is expected to be the basis of further research in the future.

## 5. Summary and Perspectives

The availability of knowledge of the molecular mechanisms of carcinogenesis, cancer inhibition, and drug action is the prerequisite for developing novel and effective clinical treatments. Understanding the role of WTAP in regulating factors, targeting pathways, and cell functions in cancer could help us develop novel clinical anticancer treatment strategies through targeting WTAP. In this review, our concentration centers on the latest research on WTAP in the cell cycle, metabolic regulation, autophagy, immune response, ferroptosis, EMT, and drug resistance. Studies have shown that WTAP knockdown regulates G1/S and G2/M phase transformation through targets such as cyclin A2 and CDK2 in cancers such as nasopharyngeal carcinoma, renal cancer, and liver cancer [35,37,38]. Under hypoxic conditions, WTAP is regulated by hypoxia inducible factor (HIF)-1α, promotes the Warburg effect of ovarian cancer cells, and significantly increases the glycolytic ability of ovarian cancer cells [45]. The WTAP research on tumor lipid metabolism is still in its infancy, which is worth further exploration in the future. According to reports, the depletion of WTAP can promote autophagy in hepatocellular carcinoma [48]. Mechanistically, WTAP regulates LKB1, which in turn mediates downstream AMPK phosphorylation, thereby promoting autophagy [48]. The immune system plays a significant role in combating cancer by identifying and destroying newly formed tumor cells through a process called cancer immunosurveillance [101]. Taku and colleagues found that WTAP plays an essential role in the growth of thymocytes. In addition, they observed that the absence of WTAP in CD4^+^ T cells resulted in reduced levels of peripheral T cells [86]. Nevertheless, the current research on the effect of WTAP in tumor immunity is mostly limited to bioinformatics analysis. Therefore, the regulatory mechanism of WTAP in tumor immunity needs further research. Furthermore, WTAP also stabilizes NRF2 expression through YTHDF1 to promote the proliferation of bladder cancer cells via inhibiting ferroptosis [51]. In hepatocellular carcinoma and ovarian cancer, WTAP depletion significantly restrains the migration and invasive ability of tumor cells [53,54]. Mechanistically, the expression of mesenchymal markers such as vimentin and Slug decreases after WTAP knockdown, indicating that the EMT process is inhibited [53,54]. In pancreatic cancer, WTAP targets the Fak-PI3K-AKT and Fak-Src-GRB2-Erk1/2 axes to promote tumor cell chemosensitivity to gemcitabine [56]. Interestingly, WTAP seems to be a double-edged sword for cancer. In CRC, carbonic anhydrase IV (CA4) inhibits tumor cell proliferation via the Wnt/β-catenin signaling pathway, while WTAP knockdown rescues the inhibition of cell viability caused by CA4 [102]. Therefore, WTAP may play complex roles in both tumor suppression and tumor promotion. Further studies are needed to fully reveal the mechanism of WTAP as a tumor suppressor gene.

## 6. Conclusions

To summarize, our review explores the fundamental mechanisms and latest discoveries of how WTAP controls cancer. The diverse functions of WTAP in the development and advancement of cancer indicate WTAP’s potential as a hopeful treatment target for cancer. Nonetheless, comprehending the precise mechanisms through which WTAP contributes to the progression and spread of tumors requires additional research, along with the production of focused therapeutic approaches that efficiently exploit its operation.

## Figures and Tables

**Figure 1 cancers-15-03053-f001:**
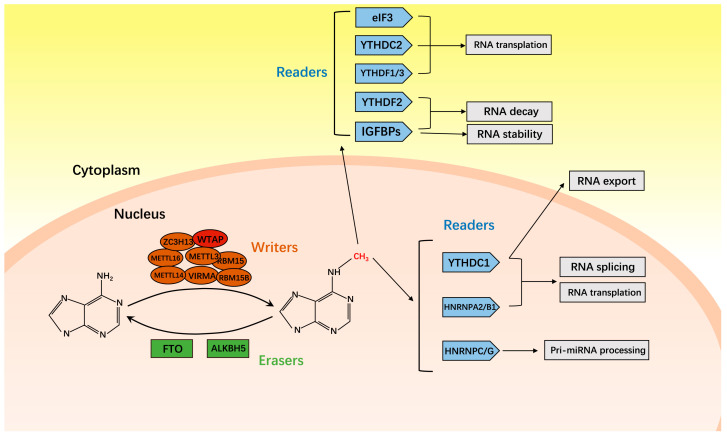
Molecular mechanism of m^6^A modification. The process of modifying m^6^A in RNA is facilitated by three key players: writers, erasers, and readers. Writers, or methyltransferases, including methyltransferase-like 3 (METTL3), METTL14, WT1-associated protein (WTAP), zinc finger CCCH-type containing 13 (ZC3H13), METTL16, vir-like m^6^A methyltransferase-associated (VIRMA), RNA-binding motif protein 15 (RBM15), and RBM15B, are responsible for the addition of m^6^A modifications onto RNA. Conversely, demethylases, referred to as erasers, work to remove m^6^A modifications. Finally, m^6^A recognition proteins, also known as readers, are tasked with recognizing m^6^A on RNA and subsequently influencing its fate, including translation, splicing, export, decay, and stability.

**Figure 2 cancers-15-03053-f002:**
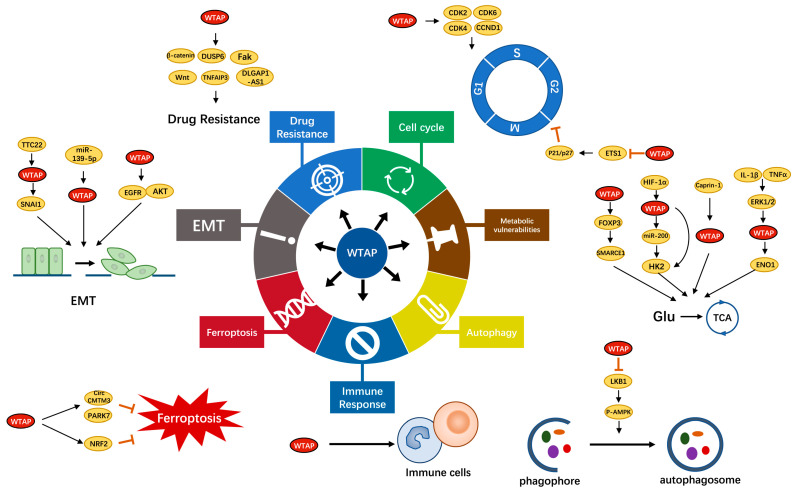
The biological function of WTAP in cancer. WTAP participates in multiple pathophysiological processes in cancer progression, including the cell cycle, metabolic vulnerabilities, autophagy, immune response, ferroptosis, epithelial mesenchymal transition (EMT), and drug resistance.

**Table 1 cancers-15-03053-t001:** Roles of WTAP in human cancers.

Function	Cancers	Regulators	Targets	Bioinformatics Research	Reference
Cell cycle and proliferative arrest	nasopharyngeal carcinoma, hepatocellular carcinoma, renal cell carcinoma, endometrial cancer	/	ETS1, p21/p27, CDK2, CDK4, CDK6, CCND1, BAX, PARP, Mcl-1	√	[35,36,37,38,39,40,41]
Metabolic vulnerabilities	colon adenocarcinoma, diffuse large B cell lymphoma, gastric cancer, ovarian cancer, esophageal carcinoma, breast cancer	HIF-1α, Caprin-1, ERK1, ERK2	FOXP3, SMARCE1, HK2, microRNA-200, ENO1	/	[42,43,44,45,46,47]
Autophagy	hepatocellular carcinoma	/	LKB1, p-AMPK	/	[48]
Immune infiltration	hepatocellular carcinoma, esophageal cancer	/	/	√	[36,49]
Ferroptosis	hepatocellular carcinoma, bladder cancer	/	circCMTM3, NRF2	/	[50,51]
EMT	colon cancer, hepatocellular carcinoma, non-small cell lung cancer, ovarian cancer, glioblastoma	TTC22, miR-139-5p	SNAI1, N-cadherin, Slug, E-cadherin, Vimentin, EGFR, AKT	/	[52,53,54,55]
Drug resistance	endometrial cancer, NK/T cell lymphoma, bladder cancer, esophageal cancer, pancreatic cancer, diffuse large B cell lymphoma, breast cancer, acute myeloid leukemia	Circ0008399, EMS, miR-758-3p, Hsp90	Wnt/β-Catenin, DUSP6, TNFAIP3, Fak, DLGAP1-AS1, miR-299-3p	/	[39,41,56,57,58,59,60,61]

## Data Availability

No new data were created or analyzed in this study. Data sharing is not applicable to be this article.

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
