# Peer review of "The Emerging, Multifaceted Role of WTAP in Cancer and Cancer Therapeutics"

_cancers, 2023, doi:10.3390/cancers15113053_

Round 1
Reviewer 1 Report
Ju et al provides a comprehensive review of the potential role of WTAP in cancer and cancer therapeutics. To this end, the authors provide an up-to-date account of WTAP’s involvement in various aspects of cancer, such as EMT, cell cycle, drug resistance, metabolism, autophagy, tumor immunology and ferroptosis. I believe that this review would benefit scientists working on epitranscriptomics in cancer biology. I have the following suggestions to improve the manuscript prior to warranting its publication.
Major points:
1. Line 47, please move the sentence beginning with “scientists have…” to a new paragraph. I would also suggest introducing the term “epitranscriptomics” rather than using the term “epigenetics” to be correct.
2. Table 1. I did not quite understand what the authors meant by “WTAP related genes”. Please provide more description of the table. Additionally, Table 1 is not cited in the main text.
3. Please repharase the lines 146-149 to improve clarity.
4. Section 3.4. WTAP and Immune Response includes a discussion on the normal T cell biology rather than tumor immunology. Please discuss how WTAP modulates tumor immunology.
5. The lines 219-220. There is ambiguity in this sentence. “…cells-cells and cytokines” is confusing.
6. Line 238, I did not quite understand why authors specifically included ferroptosis only out of so many different types of cell death. For example, why not cuproptosis, parthanator or necroptosis?
Minor points:
1. Line 16, please use “deposition” instead of “installation”; line 16, please use “RNA methyltransferases” instead of “methyltranferases” .
2. Line 59, please put space right after “enzymes”.
3. Line 84, capitalize “mettl3”.
4. Line 110 is a repetition of line 77. Please skip this sentence.
5. Line 114, place a period after “resistance”.
6. Line 162, use “Additionally” instead of “And”. The same goes for “And” in line 188.
7. Line 168, “Glucose” should be “glucose”.
8. Line 191, “clone” should be “clonal”.
9. There are several uses of “mechanically”, which I believe should be “mechanistically”. Please check the whole manuscript.
10. Line 285, “Pathway” should be “pathway”.
11. Line 326, “Summarize and Prospective” should be “Summary and Perspectives”.
12. Line 347, “affect” should be “effect”.
I believe that the manuscript could greatly benefit from proofreading by a Native speaker.
Reviewer 2 Report
This review “The Emerging, Multifaceted Role of WTAP in Cancer and Cancer Therapeutics” by Ju et al is devoted to summarizing the role of WTAP in human cancers and the relationship between WTAP1 and m6A. This topic is very intriguing and interesting. I recommend to undertake more attentive work with writing to improve the manuscript before it will be published.
For part 3” Function and role of WTAP”, I recommend using figures showing the signaling pathways and molecular interactions to increase comprehension, and especially highlight the newly findings of WTAP functions.
The text of the manuscript has some mistakes that complicate the perception and analysis of information.
Reviewer 3 Report
Dear Authors,
the topic is interesting and translational.
Please, address the following issues:
1. Line 48: m1A and m5C should be written in extenso the first time.
Line 51: "proved to be present..."
3. The lines 52-54 should be rephrased because English is not understandable.
4. The period between lines 56-62 is too long and results confused. A recent and clear overview of the three different classes of enzymes directly involved in m6A modification is published in: "Cusenza VY et al. The lncRNA epigenetics: the significance of m6A and m5C lncRNA modifications in cancer. Frontiers Oncol 2023 PMID: 36969033". This manuscript should be commented and cited.
5. Figures: please, add some more artwork representing the key pathways correctly described in paragraphs 3.1 - 3.7
The English is fine, except for some sentences (see the comments above) that need to be rephrased.
Minor spelling errors should be checked.
